# Methodology for Determining the Location of Intermodal Transport Terminals for the Development of Sustainable Transport Systems: A Case Study from Slovakia

**Ján Ližbetin**

Department of Transport and Logistics, Faculty of Technology, Institute of Technology and Business in České Budějovice, 37001 České Budějovice, Czech Republic; lizbetin@mail.vstecb.cz; Tel.: +420-387-842-190

**Abstract:** A high-quality infrastructure and technical base is a vital factor in the development of intermodal transport in transport systems. Intermodal transport terminals are the most important component of a combined transport infrastructure, providing an essential connection between different modes of transport. This article deals with the issue of where to locate intermodal transport terminals within a transport network. In reality, this decision comes down to the potential of a particular location (e.g., an industrial park) and the critical role of private investors. These are mostly subjective factors, whereby little or no consideration is given to objective criteria. Within this context, it is extremely important that decisions are taken with regards to the development and construction of public networks, and economically neutral intermodal transport terminals by independent subjects are based on a non-discriminatory approach. In other words, it is essential that such terminals are built in places that comply with the stated priorities of the transport policy of a specific state. In this article, the author puts forward a method for determining the location of terminals that are based on the optimisation of several influential factors. The specified methodology is applied to a case study in Slovakia. The theoretical part of the article deals with the nature of the method to be applied. The discussion part involves a case study concerning the (potential) location of intermodal transport terminals in the Slovak Republic.

**Keywords:** intermodal transport; intermodal transport terminal; location; sustainability; case study

## 1. Introduction

The importance of the issue addressed is indisputable. The sustainability of a transport system in any state or a unit (e.g., the EU) consists in a harmonized use of available transport infrastructures capacities, the completion of several smaller shipments into a bigger one, and the transportation of goods for longer distances using modes of transport with greater capacity. This objective is achievable by operating intermodal transport chains. To ensure that intermodal transport is attractive and evolving, it is necessary to build quality and available infrastructure. Intermodal transport terminals are the most necessary element of transport infrastructure for intermodal transport. The terminal location and capacity depend on their future use and intermodal transport development.

Intermodal transport terminals (ITTs) are one of the most essential components of combined transport infrastructure, providing a vital connection between individual modes of transport. They represent a systemic point, at which the mode of transport (rail, water, road) and the handling of intermodal transport units (large containers, swap bodies, road semi-trailers, and transport units), change [1].

Within the European context, in particular with regards to the situation in countries of the Visegrad group, and in this case in the Slovak Republic (SR), one of the main reasons for the slower development of combined transport is the existing number, condition, and technical facilities of the ITTs situated nationwide [2]. With the exception of a new terminal in Žilina–Teplička, these terminals are basically private trans-shipment points for container transport (onward transport of containers from seaports to inland points) with non-discriminatory access to customers. In addition, the majority of the terminals were built in the 1980s (i.e., before the official establishment of the SR) when their primary operations corresponded to the original concept of progressive transport systems, namely the conveyance of large twenty-foot containers. Furthermore, no economically neutral terminals focused on continental transport systems (transport of swap bodies and road semi-trailers) have been constructed [3,4], which is a highly restrictive factor in the development of a combined transport infrastructure in the SR.

In 2001, a nationwide combined transport development project [5] foresaw the need for the construction of 13 ITTs, of which 5 should be of international importance. Assuming the SR covers an area of 49,034 km$^2$, this equates to approximately one terminal per 3772 km$^2$, which is slightly more than in Austria and almost twice the density in Germany [6].

The objective of the contribution is to propose a methodology of locating intermodal transport terminals so that the existing transport infrastructure is effectively used and the potential of using terminals is considered. The methodology also considers the complex coverage of the area. An accessible and functioning intermodal transport terminal is a necessary condition for the development of a sustainable intermodal transport system.

The proposed methodology offers an exact solution for terminals placement in the Slovak Republic, taking into account the relevant factors that could affect the future use of the terminal. Of course, the methodology is applicable to any state.

## 2. Literature Review

The European Agreement on Important International Combined Transport Lines and Related Installations (AGTC) is the underlying document that defines combined transport in Europe [7]. Annex II thereof contains a list of objectivess (terminals) in relation to the participating countries. Those listed in the SR are historically linked to container trans-shipment points constructed by Czechoslovak Railways. Annex IV sets out the technical and technological requirements for the listed terminals, whereby one of the requirements is their economically neutrality, i.e., non-discriminatory access of terminal operators to individual customers. Unfortunately, none of the operational terminals in Slovakia currently fulfil this requirement.

The indisputable importance of intermodal transport for securing a sustainable transport system is noted e.g., in the contribution [8]. The author points to the environmental advantages of intermodal transport chains, the necessary cooperation between the individual modes of transport, and the importance of intermodal transport terminals accessibility. Those are the factors most affecting the sustainability and development of the transport system.

In the paper [9], the authors state that, confidence in intermodal transport has not yet been defined. There are many different approaches to the concept of trust. However, the authors have integrated them with the paradigm shift in light of the challenges of sustainability.

The issue of the ITTs and their location has been addressed by several authors [10–16]. An interesting perspective on the location of a terminal is put forward by [10], whereby the most suitable location is determined in relation to the various parties involved (investors, users, local administration and inhabitants). Such parties often have conflicting goals and interests, which makes it necessary to define a large number of criteria for the overall evaluation. As a result, the authors propose a new hybrid Multiple-criteria decision analysis (MCDM) model that combines Fuzzy Delphi Methods in order to support the decision making process. They add that the proposed model was developed in a fuzzy environment (in order to overcome the ambiguities and conflicts in, and between, evaluation criteria, sub criteria, and decision alternatives) and applied to the terminal located in Belgrade.

Unfortunately, this approach neither solves the need for a comprehensive network of terminals nationwide, nor does it ensure serviceability of an entire territory, but merely determines the need for the terminal in the location of interest.

A specific methodology for locating ITTs in Croatia is discussed by [11]. The authors base their findings on relevant qualitative criteria (indicators) for intermodal terminals: Flexibility, safety, reliability, time, and availability. On the grounds of such indicators, and in accordance with European transport policy, the authors further developed and evaluated the following localisation criteria: Legislative, environmental, commodity flow, spatial, technical, technological, and organisational. These criteria were subsequently sub-divided and evaluated, using the Analytical Hierarchy Process (AHP). The results show that the commodity flow criterion (followed by the spatial criterion) is the most influential when it comes to selecting a particular end-point. The author therefore considers this criterion to be decisive, but goes on to include other preferred evaluation criteria in his proposed model.

Although another paper, in [12], points out the significance of the application of mathematical solutions to locational problems and examines several mathematical models, as defined in [13,14], it does note their considerable complexity and the time-consuming nature thereof.

It is within this context that this article sets out a new programming model that generates quicker results. However, it does not discuss the input data for the model, but only the calculation process itself.

## 3. Materials and Methods

### 3.1. Theoretical Basis for Developing a Mathematical Model for Determining the (Al)Location of ITTs

In the past, the strategies employed to determine the number and location of ITTs in the SR, followed by the examples of other European countries, and were also influenced by factors, such as their likely potential and their even geographical distribution across the SR [17]. However, such decisions tended to be biased and based on subjective criteria, and not supported by exact methods, for example, through the inclusion of optimisation criteria in the decision-making process. At present, several mathematical and optimisation methods exist for determining the appropriate (optimal) location of terminals on the basis of defined optimisation criteria [18,19].

The main issue of the location of intermodal transport terminals can generally be equated to that of determining the location of service centres within a given network. It can be argued that the total transport capacity for collections and deliveries by road in terms of intermodal transport units is dependent on the number and location of ITTs. However, the optimisation criterion here is to minimise the overall transport capacity for collections and deliveries (expressed in thousands of tonne-kilometres, i.e., tkm). This implies that combined transport potential (expressed in thousands of tonnes) must be taken into account. Consequently, putting forward a proposal for a specific method, to determine the optimal number and location of terminals, is a complex task, which requires a number of factors underlying the choice of terminal location to be simultaneously considered.

While developing a mathematical model for determining the location of ITTs, it is essential to take into consideration the level of potential in their service areas. Existing terminals, trans-shipment points and transport infrastructure must therefore be included in such an assessment, which implies that terminals should be built in areas with good road and railway infrastructure links.

Within the context of this study, the location of the ITTs within the given transport network will be determined on the basis of the road network, which is based on the need to minimise the costs of collections and deliveries within the service areas under consideration. With respect to the extent and solution of the task, as well as the required quality of the road infrastructure, the full length and breadth of the road network is not taken into consideration. More specifically, this has been limited to 1st and 2nd class roads and important 3rd class roads that connect them. The actual network is represented by a graph structure, whereby the final set of significant points are referred to as vertices and the final set of lines joining (some of) those vertices, and which represent communication links, are referred to as edges. The simplest graph structure is one generated with non-oriented edges (graph) [20].

The vertices in this network are formed by road junctions, road endings or other important points, particularly points (places), where possible transport potential can be found, or created, i.e., mostly in the vicinity of large or major towns (cities), or directly inside them. Owing to these conditions, essentially all district towns (on assuming the existence and/or possible future construction of potential resources for combined transport) and some other towns (given their geographical location) are taken into consideration.

Each vertex in the network is assigned a non-negative number (the so-called vertex weight) which expresses the significance of the given vertex [21]. Each edge, which represents a road section, linking two adjacent vertices, is also expressed as a non-negative number that indicates the length of the given edge, i.e., the distance in kilometres between the two adjacent vertices.

### 3.2. Construction of Initial Matrix for Multi-Criteria Evaluation

The main purpose of the mathematical model, when taking a certain (final) number of (theoretically appropriate) vertices into consideration, is to select those vertices that are most suitable for locating an intermodal transport terminal. To be able to evaluate the vertices in relation to the above factors, which influence the choice of location at a given vertex, and which should be considered while designing the methodology, it would appear to be advantageous to use a group of multi-criteria evaluation methods [22].

In general terms, multi-criteria evaluation methods can be used to compare and select any objects (terminals), based on a number of indicators. Due to their ability to synthesise different indicators (characteristics) into a quantitative composite indicator, such methods are particularly suitable for analysing an object's (née terminal's) position in the market (in a network). Additionally, they allow one to compare a set of objects based on their characteristic activities, and, at the same time, to determine a location order for the analysed objects. Their selection is subsequently determined by the location´s place in the order after taking all the characteristics into account [23].

The basis for multi-criteria evaluation is an initial matrix. In this case, the matrix consists of objects and their characteristics, whereby the objects represent suitable places (vertices) for locating ITTs, and the characteristics are expressed by technical and technological indicators. Correspondingly, indicators of places (vertices) that may significantly affect the location of ITTs (i.e., general ITT location criteria) are compared as well.

When constructing the initial matrix of objects, it is advisable to go through the following steps:

1. Select the objects to be included in the analysed set.
2. Select indicators that characterise the objects´ activities.
3. Determine the weights for each of the indicators.
4. Develop an initial matrix (see Table 1).

**Table 1.** Development of initial matrix of objects.

| Object/Indicator | $X_1$ | $X_2$ | ... | $X_m$ |
|---|---|---|---|---|
| 1 | $X_{11}$ | $X_{12}$ | ... | $X_{1m}$ |
| 2 | $X_{21}$ | $X_{22}$ | ... | $X_{2m}$ |
| ... | ... | ... | ... | ... |
| N | $X_{n1}$ | $X_{n2}$ | ... | $X_{nm}$ |
| weights of indicators | $P_1$ | $P_2$ | ... | $P_m$ |

Source: Author.

In Table 1:

$X_{ij}$—is the value of the *j*-th indicator in the *i*-th object;

*m*—is the number of indicators;

*n*—is the number of evaluated objects;
*p_j*—is the weight

The weight ($p_j$) may be:

- normalised (thereby fulfilling the requirement of $0 \le p_j \le 1$; $j = 1, 2, \ldots m$; where $p_1 + p_2 + \ldots \ldots \ldots + p_{m-1} + p_m = 1$); or
- differentiated (integers expressing the weights of the individual criteria).

As previously mentioned, the main objective of multi-criteria evaluation methods is to transform and synthesise the values of various indicators into a single composite indicator (resulting characteristics), thereby expressing the position of a particular individual object within the set of objects under consideration. Due to the fact that none of the multi-criteria evaluation methods adequately capture the specifics of the topic under research, it is appropriate to use a partially adjusted method of the weighted sum of order, in combination with evaluations of the characteristics using normalised weights. The overall level (significance) of a given vertex, i.e., suitability of an ITT location, is then expressed by the composite indicator [24].

This indicator can be defined as a composite indicator, according to which a certain weight (significance) is assigned. It is calculated as the sum of the product weights of individual criteria and the selected criteria indicators:

$$K_i = \sum_{j=1}^{m} p_j * s_{ij} \tag{1}$$

where:

$K_i$—is the coefficient of the *i*-th vertex, $i = 1, 2, \ldots, n$;
*n*—is the number of vertices in the network;
$p_j$—is the normalised weights of individual criteria, $\sum p_j = 1$,
*m*—is the number of criteria;
$s_{ij}$—is the selected criteria indicator.

It should be noted that the aforementioned indicator expresses the significance of the vertex being examined, which will have a major influence on the decision about the location of a terminal at the given vertex. It is therefore relevant to select such characteristics (indicators) that best reflect the specifics of the situation at the given vertex, i.e., to take into account all the factors that influence and determine if an ITT should be located there [25].

### 3.3. Mathematical Model

The issue at hand is to find answers to two underlying questions: What is the optimal number of terminals? Where should they be located? Simply, this represents a location-allocation dilemma. Location-allocation dilemmas seek a response to two questions at the same time. How many terminals in the network (allocation)? Where to place the terminals in the network (location)?

Methods and algorithms exist that seek to answer both these questions at the same time, for example, the Branch and Bound Method, and methods that can only find either the optimal number of terminals (Continuous Approximation Method) [26], or the optimal location of terminals when their number is assigned (Iterative Method) [27]. Which method should be used depends on the accuracy of the required solution, the availability of time, and the input data. Given the current nature of the issue at hand, and the specified optimisation criterion, the Iterative Method would appear to be adequate in this respect, as has been suggested by a number of other authors [28,29]. Although this method is relatively simple, it provides basic information on the suitability of proposed ITT locations if their number is specified.

The optimal number of terminals may also be obtained using the so-called "approximation" method, i.e., by gradually increasing the number of terminals while solving an allocation task. Individual

variants are then compared, on the basis of the comprehensive coverage of the area under consideration, and the size of the service area of each terminal. Within this context, the decisive criterion for determining the number of terminals is the maximum distance that needs to be covered by road transport for tax exemptions for motor vehicles (sometimes referred to as road tax) to be applied to combined transport users. In the SR, this is legally embodied in the Motor Vehicle Tax Act No. 361/2014 Coll., which, for the purposes of defining combined transport refers to Act No. 514/2009 Coll., on Rail Traffic, states that the maximum distance for collections and deliveries by road is 150 km. As a result, the optimal number of terminals is conditioned by the comprehensive coverage of their relevant service areas and the maximum distance by road between each terminal, i.e., 150 km.

The allocation task can be solved through the application of graph theory. From the mathematical viewpoint, the minimum value for the criterion function should be found. The proposed solution for the allocation task (relationships, algorithms) was developed according to [30–34], and is thoroughly described in [35].

The criterion function for locating the intended terminal points within a particular network is defined as follows: A certain $k$ set of $D_k$ terminal points ($|D_k| = k$) shall be viewed as the vertex optimal location of the $k$ terminal points within the $G = (V, H)$ network provided that:

$$f(D_k) = \min_{D_k}\{f(D_k)\} \tag{2}$$

where:

$D_k$—is all $k$-element subsets of $V$ vertices and:

$$f(D_k) = \sum_{d \in D_k} \sum_{u \in A(d)} 2 * d(d,u) * w(u) \tag{3}$$

where:

$A\check{}(d)$—is the allocated service area of the $d$ terminal point;
$d(d,u)$—is the operations of the $u$ vertex from the $d$ terminal point;
$w(u)$—is the weight of the $u$ vertex (composite indicator).

The criterion functions were defined in order to express the amount of traffic involved in the network operations (collections and deliveries by road). Return journeys to terminals are assumed to take place along the same routes, in which case the terminals (terminal points) may be only placed at vertices [36].

In terms of combinatorics, the task discussed here is of the $Ck(p)$ type [31]. It is necessary to determine a certain combination of the $k$-th class of $p$ elements for which the criterion functions are minimised. Using the well-known Hakimi algorithm [37,38], the optimisation process for locating terminal points (intermodal transport terminals) within the given network can be solved by applying an iterative algorithm. However, this algorithm assumes assigning a distance matrix, which requires the determination of the minimum distances between all vertices and the generation of a network distance matrix. In order to determine such distances, the shortest path routing algorithm can be used in accordance with [39].

The iterative algorithm computation method using the Hakimi algorithm is shown schematically in Appendix Figure A1.

## 4. Results

In the previous section, the process of developing the mathematical model for determining the allocation of ITTs was theoretically described. Following from the theoretical basis, the proposed model was applied to the existing conditions in the SR.

*4.1. Construction of Initial Matrix*

The first step was to construct an initial matrix for determining the weights of the individual vertices. The creation of the designated network, as well as the definition of the vertices and edges were described above.

The particular indicators influencing the location of ITTs at given vertices and included in the initial matrix were as follows.

4.1.1. Potential of Combined Transport

This indicator may be considered as decisive in terms of selecting suitable locations for ITTs. The potential of combined transport consists in a potential amount of goods transported by combined transport (containers, swap bodies, and road semi-trailers). This potential also represents the economic potential of the area where the terminal is to be constructed. However, it is unrealistic to identify combined transport potential for each particular vertex in the given network. In various statistics and studies [5], this type of potential is sub-divided into individual regions, which implies that "regional potential" must be appropriately subdivided between individual vertices (towns, cities) situated in a single region. Furthermore, each vertex must be assigned a certain coefficient that expresses the contributory level of potential to the region's total potential, whereby the coefficients can be determined according to several possible indicators. The most apposite indicator could be one that describes the extent to which a given vertex (town, city) is involved in total production and regional production (industry, agriculture, and transport). This is reflected in the Gross Domestic Product (GDP) [40] of that vertex. Gross domestic product represents a relevant economic potential of the area analysed, having a significant influence on the location and effective operation of terminals. However, for the case study dealing with the location of intermodal transport terminals in Slovakia, this indicator cannot be used, since there are no statistical data of the share of individual vertices (towns) on the region's overall gross domestic product. If there are such statistics available (when applying the methodology in other state), this indicator is considered the most relevant.

Another indicator, which is not quite accurate (yet sufficiently quantifiable) is that of the population of the given vertex. Theoretically, it may be assumed that vertices (towns, cities) with larger populations will be more productive and will show higher consumption and demand for transport, hence holding greater combined transport potential [41,42]. This suggests that each vertex (town, city) should be assigned a certain coefficient based on a percentage of the region's population, with the total population being the sum of the populations of the vertices in the region. Such a coefficient can subsequently be used to multiply the respective regional potential. The resulting coefficients as well as the combined transport potential of each vertex are listed in Table A1.

4.1.2. Existing Terminals (Trans-Shipment Points) for Combined Transport

When determining the location of new ITTs, account must be taken of the existing terminals and trans-shipment points, in particular public ITTs that have been built and included in the AGTC. Similarly, the possible location of terminals on so-called "greenfield sites" should be disregarded.

4.1.3. Links to Railway Infrastructure

A basic pre-requisite for the location of ITTs is a good connection to the railway infrastructure. Not all railway lines and stations were taken into consideration. The focus was on those already included in the AGTC and those considered for inclusion [7].

4.1.4. Links to Road Infrastructure (Important Road Junctions)

In terms of combined transport and its transport systems, road transport performs a specific "collection" function, i.e., the collection of the entire load in the service areas of the terminals and the provision of sufficient potential for the creation of block (unit) trains [43]. It is therefore essential

for ITTs to be located at road junctions, within the service areas, whereby every aspect of terminal operations should be sufficiently accessible to all end points.

### 4.2. Determination of Indicator Weights

The determination of the weights for the relevant indicator(s), as a means by which to express their significance, is a relatively subjective matter. It should be noted that multi-criteria evaluation methods tend to organise evaluated objects according to high-weight indicator values [44]. The first indicator, i.e., the potential of combined transport, is the most important indicator affecting the choice of terminal locations. It is therefore defined as the main ("central") indicator, the one that the other indicators only modify with their weights. Establishing the central indicator is also conditioned by the fact that any vertex that has great potential, but that does not comply with the other selected indicators, would be at a certain disadvantage. Furthermore, determining the weights of the other indicators would be another problem because they are combinations of value and natural indicators [45].

In view of the above, it is appropriate for the central indicator, i.e., the potential of combined transport ($s_{pot}$), to be assigned a combined weight of ($p_1$) = 1.

The other indicators subsequently modify the central indicator with their normalised weights:

- For trans-shipment points ($s_{tsp}$) a weight of ($p_2$) = 0.3;
- for links to railway infrastructure ($s_{rwi}$) a weight of ($p_3$) = 0.2;
- for links to road infrastructure ($s_{roi}$) a weight of ($p_4$) = 0.5 (the most important in terms of shipments);

where $\Sigma p_j$ = 1, for $j$ = 2, 3, 4.

### 4.3. Construction of Initial Matrix and Development of Mathematical Model

Based on the above methodology, an initial matrix was constructed and the cumulative coefficients for the individual vertices calculated (see Table A2). The initial matrix then served as a basis for calculating the service areas of the individual ITTs, whereby the maximum distance of the vertices and terminals by road was observed. The calculations were made using the algorithm described in Section 3.2.

## 5. Discussion

When considering an option with five terminals, 80% of the total potential is located within their service areas at a distance of up to 150 km (this also being the maximum distance for road transport in terms of tax relief for motor vehicles). With six ITTs, there is 85% potential within the same distance, with 7 terminals this stands at 95%, and with eight ITTs, all relevant vertex are located within 150 km of the nearest terminal, as illustrated in Appendix Figure A2 below.

Using mathematical optimisation methods, it can be stated that the research results essentially confirmed the original concept for terminal locations [5], and identified other possible areas for the location of new ITTs. The towns of Trnava and Nitra hold great potential, with large industrial parks (and a new terminal in Nitra) being built there.

In Central Slovakia, the construction of an intermodal transport terminal in Zvolen is being considered after its location was confirmed under an allocation programme.

The location of the intermodal transport terminals should be decided by the state by means of the Ministry of Transport. In terms of European directives (COUNCIL DIRECTIVE on the development of the Community's railways 91/440/EEC), the intermodal transport terminals are a part of railway transport infrastructure. Access to the infrastructure must be non-discriminatory. The construction and development of the infrastructure is also the state competence.

The construction and operation of intermodal transport terminal networks can be ensured by the private sector, which is fully responsible for deciding whether to build the terminal in a given location, but is required to operate the terminal in terms of the aforementioned directive, that is, in a

non-discriminatory way, under economically neutral conditions. Of course, it is still about the network of, so-called, public and open intermodal transport terminals.

The objective of the intermodal transport terminal operators is to build a quality transport infrastructure in order to ensure the sustainable growth of transport system in the state, or its territory.

It would be possible to evaluate the above options from the economic standpoint as well, but this would require a detailed analysis. Such an analysis would be complicated by the difficultly associated with defining the operating costs of individual ITTs. This is because each of them would operate in a different way, and is likely to show varying levels of performance or output. Despite this, it could be argued that a larger number of ITTs would possibly generate greater demand for their services, thereby increasing the total potential of combined transport. However, an economic evaluation was not the objective of this study.

There are also non-economic effects that need to be taken into consideration when it comes to the construction of new ITTs. Developed and high-quality transport infrastructure is a pre-requisite for an increase in combined transport in relation to the total transport, irrespective of whether this concerns import, export, or transit. This reduces the unfavourable effects of road freight transport, emissions, accident rates, congestion on international corridors, thereby ensuring the sustainability of the transport system.

Finally, locating ITTs must be carried out sensibly (with regards to the efficiency and use of each terminal), whereby it is necessary to be able to respond flexibly to the ever-changing transport market. Evidence of this approach is the creation of large industrial parks in a number of locations across the SR, which form potential sources for combined transport as well (e.g. Nitra). Unfortunately, the SR did not adequately respond to these sources in the past, which resulted in the establishment of private (non-public) trans-shipment points for combined transport (Púchov, Sládkovičovo, Dunajská Streda). The top priority of the transport policy of the SR should, therefore, be the construction and modernisation of terminals in areas with such transport potential, so that future carriers can make full use of the public terminals and their services. As a result, they would not be forced to build their own non-public trans-shipment points that are not in accordance with the AGTC and would have access to services based on non-discriminatory practice or a economically neutral approach.

This methodology is focused on locating intermodal transport terminals (hubs for intermodal transport units between the individual modes of transport). Intermodal transport terminals are basic (central) elements of logistic centres, and gateway terminals. The issue in locating public logistic centres are thus closely related with the location of intermodal transport terminals. This practice shows that the construction of an intermodal transport terminal is often followed by its extension by various warehouses and distribution centres. The existence of a terminal (available transport infrastructure) causes a synergistic effect in the form of further warehouses construction, thus extending the services provided by the terminal by other logistic services. The methodology, however, is focused on the substantial part of such centres—terminals, whose construction supports the development of sustainable transport systems in the area. Intermodal transport terminals must be compatible with all transport systems (which means being able to adapt to transhipment of sea containers, swap bodies, or intermodal road semi-trailers).

## 6. Conclusions

In the past, the concept of placing terminals was not based on any exact methodology. In Slovakia, for example, the concept of placing terminals was based on the example of terminals placing in Germany, and no criteria and factors affecting the proper terminal location were taken into consideration. Some current methodology for designing the concept of placing terminals are described in Chapter 2 of the contribution; however, none of them is based on the factors that the proposed methodology includes. The basic idea of the methodology (and its minimalist function) is to minimize the movement of road vehicles on the road, when collecting and delivering consignment within the Terminal Catchment Area. The reason is to minimize the negative effects of road freight transport in terms of external costs. Moreover, a "bonus" for road hauliers, in the form of road tax relief, is considered.

The characteristics of the proposed methodology thus include:

(a)   Creating a graph structure consisting of a road network.
(b)   Assigning importance weights to individual vertices (when taking into consideration the criteria influencing the placement choice).
(c)   Calculating the Terminal Catchment Area of the individual terminals when taking into consideration the minimization of road vehicles journeys, as well as the kilometre distance of the terminal 150 km by road (for a possible road tax relief).

Such methodology has not been published and can, therefore, be described as a new/different view of the methods for placing intermodal transport terminals.

The next step in the future should be solving the financial issue of the construction of intermodal transport terminals. The construction could be supported, for example, by Operational Program Transport.

**Supplementary Materials:** The following are available online at http://www.mdpi.com/2071-1050/11/5/1230/s1, Figure S1: Iterative algorithm flowchart, Table S1: Determination of coefficients and potential of individual vertices, Table S2: Initial matrix of objects.

**Funding:** This research received no external funding.

**Acknowledgments:** The author would like to thank members of staff at the Ministry of Transport and Construction of the Slovak Republic for the statistical data provided.

**Conflicts of Interest:** The author declares no conflict of interest.

## Appendix A

Appendix Figure A1 contains an iterative algorithm flowchart. The algorithm was used to develop a mathematical model for locating intermodal transport terminals.

Table A1 and Table S1 contains a table of the potential of individual vertices determined by converting the potential of the respective region.

**Table A1.** Determination of coefficients and potential of individual vertices.

| Vertex | Potential of Region | Population | Potential of Vertex | Coefficient |
|---|---|---|---|---|
| Bratislava | 22,819 | 425,533 | 202,495,806 | 0.8874 |
| Malacky | 22,819 | 17,870 | 8,511,487 | 0.0373 |
| Kúty | 9134 | 4200 | 1,653,254 | 0.0181 |
| Holíč | 9134 | 11,560 | 4,557,866 | 0.0499 |
| Skalica | 9134 | 14,980 | 5,909,698 | 0.0647 |
| Senica | 9134 | 21,061 | 831,194 | 0.091 |
| Pezinok | 22,819 | 21,077 | 1,004,036 | 0.044 |
| Senec | 22,819 | 15,030 | 7,142,347 | 0.0313 |
| Dunajská Streda | 9134 | 23,518 | 928,0144 | 0.1016 |
| Trnava | 9134 | 69,488 | 27,420,268 | 0.3002 |
| Galanta | 9134 | 16,019 | 6,320,728 | 0.0692 |
| Sereď | 9134 | 17,317 | 683,2232 | 0.0748 |
| Hlohovec | 9134 | 23,264 | 917,967 | 0.1005 |
| Piešťany | 9134 | 30,066 | 1187,42 | 0.13 |
| Nové Mesto nad Váhom | 7546 | 20,976 | 6,089,622 | 0.0807 |
| Myjava | 7546 | 12,924 | 3,750,362 | 0.0497 |
| Trenčín | 7546 | 57,051 | 16,555,924 | 0.2194 |
| Dubnica nad Váhom | 7546 | 25,741 | 747,054 | 0.099 |
| Ilava | 7546 | 5412 | 1,569,568 | 0.0208 |
| Púchov | 7546 | 18,654 | 5,410,482 | 0.0717 |
| Považská Bystrica | 7546 | 42,490 | 12,330,164 | 0.1634 |
| Bytča | 8901 | 11,495 | 3,248,865 | 0.0365 |
| Žilina | 8901 | 85,278 | 24,112,809 | 0.2709 |
| Kysucké Nové Mesto | 8901 | 16,526 | 4,673,025 | 0.0525 |

**Table A1.** *Cont.*

| Vertex | Potential of Region | Population | Potential of Vertex | Coefficient |
|---|---|---|---|---|
| Čadca | 8901 | 26,443 | 747,684 | 0.084 |
| Prievidza | 7546 | 52,070 | 15,114,638 | 0.2003 |
| Partizánske | 7546 | 24,686 | 71,687 | 0.095 |
| Topoľčany | 8810 | 28,819 | 826,378 | 0.0938 |
| Nitra | 8810 | 86,138 | 2,468,562 | 0.2802 |
| Šurany | 8810 | 10,415 | 298,659 | 0.0339 |
| Nové Zámky | 8810 | 41,669 | 1,193,755 | 0.1355 |
| Šaľa | 8810 | 24,514 | 702,157 | 0.0797 |
| Komárno | 8810 | 36,804 | 1,054,557 | 0.1197 |
| Hurbanovo | 8810 | 8055 | 230,822 | 0.0262 |
| Štúrovo | 8810 | 11,410 | 326,851 | 0.0371 |
| Levice | 8810 | 36,476 | 1,044,866 | 0.1186 |
| Šahy | 7121 | 8059 | 1,737,524 | 0.0244 |
| Vráble | 8810 | 9501 | 272,229 | 0.0309 |
| Zlaté Moravce | 8810 | 13,646 | 391,164 | 0.0444 |
| Žarnovica | 7121 | 6543 | 1,409,958 | 0.0198 |
| Žiar nad Hronom | 7121 | 19,741 | 425,1237 | 0.0597 |
| Banská Štiavnica | 7121 | 10,873 | 2,342,809 | 0.0329 |
| Hontianske Nemce | 7121 | 1495 | 320,445 | 0.0045 |
| Krupina | 7121 | 7847 | 1,687,677 | 0.0237 |
| Zvolen | 7121 | 43,488 | 9,364,115 | 0.1315 |
| Hronská Dúbrava | 7121 | 391 | 85,452 | 0.0012 |
| Banská Bystrica | 7121 | 81,961 | 17,652,959 | 0.2479 |
| Turčianske Teplice | 8,680 | 6943 | 191,828 | 0.0221 |
| Martin | 8901 | 59,772 | 16,902,999 | 0.1899 |
| Vrútky | 8901 | 7242 | 204,723 | 0.023 |
| Dolný Kubín | 8901 | 19,855 | 5,616,531 | 0.0631 |
| Kraľovany | 8901 | 472 | 133,515 | 0.0015 |
| Námestovo | 8901 | 8126 | 2,296,458 | 0.0258 |
| Tvrdošín | 8901 | 9464 | 2,679,201 | 0.0301 |
| Ružomberok | 8901 | 30,166 | 8,527,158 | 0.0958 |
| Liptovský Mikuláš | 8901 | 32,966 | 9,328,248 | 0.1048 |
| Brezno | 7121 | 22,573 | 4,863,643 | 0.0683 |
| Detva | 7121 | 15,024 | 3,232,934 | 0.0454 |
| Lučenec | 7121 | 28,146 | 6,059,971 | 0.0851 |
| Veľký Krtíš | 7121 | 13,988 | 3,012,183 | 0.0423 |
| Slovenské Ďarmoty | 7121 | 545 | 113,936 | 0.0016 |
| Fiľakovo | 7121 | 10,271 | 2,214,631 | 0.0311 |
| Rimavská Sobota | 7121 | 24,810 | 534,075 | 0.075 |
| Poltár | 7121 | 6035 | 1,303,143 | 0.0183 |
| Hnúšťa | 7121 | 7560 | 1,630,709 | 0.0229 |
| Tornaľa | 7121 | 8022 | 1,730,403 | 0.0243 |
| Rožňava | 9372 | 19,130 | 4,573,536 | 0.0488 |
| Revúca | 7121 | 13,273 | 2,855,521 | 0.0401 |
| Poprad | 7451 | 55,680 | 11,966,306 | 0.1606 |
| Spišská Nová Ves | 9372 | 38,785 | 9,268,908 | 0.0989 |
| Levoča | 7451 | 14,511 | 3,121,969 | 0.0419 |
| Spišský Štvrtok | 9372 | 2273 | 543,576 | 0.0058 |
| Kežmarok | 7451 | 12,798 | 274,9419 | 0.0369 |
| Stará Lubovňa | 7451 | 16,398 | 3,524,323 | 0.0473 |
| Sabinov | 7451 | 12,328 | 2,652,556 | 0.0356 |
| Bardejov | 7451 | 33,402 | 7,182,764 | 0.0964 |
| Svidník | 7451 | 12,392 | 2,667,458 | 0.0358 |
| Prešov | 7451 | 92,147 | 1980,4758 | 0.2658 |
| Spišské Podhradie | 9372 | 3855 | 918,456 | 0.0098 |
| Margecany | 9372 | 2035 | 487,344 | 0.0052 |
| Gelnica | 9372 | 6243 | 1,490,148 | 0.0159 |

**Table A1.** *Cont.*

| Vertex | Potential of Region | Population | Potential of Vertex | Coefficient |
|---|---|---|---|---|
| Moldava nad Bodvou | 9372 | 9685 | 2,314,884 | 0.0247 |
| Košice | 9372 | 235,281 | 56,232 | 0.6 |
| Trebišov | 9372 | 22 765 | 543,576 | 0.058 |
| Slovenské Nové Mesto | 9372 | 1073 | 253,044 | 0.0027 |
| Michalovce | 9372 | 39,915 | 9,540,696 | 0.1018 |
| Sobrance | 9372 | 6317 | 149,952 | 0.016 |
| Dobrá pri Č.n.T. | 9372 | 395 | 9372 | 0.001 |
| Vranov nad Topľou | 7451 | 23,020 | 4,947,464 | 0.0664 |
| Strážske | 9372 | 4457 | 1,068,408 | 0.0114 |
| Humenné | 7451 | 35,043 | 7,532,961 | 0.1011 |
| Snina | 7451 | 21,390 | 4,597,267 | 0.0617 |
| Medzilaborce | 7451 | 6699 | 1,438,043 | 0.0193 |
| Stropkov | 7451 | 10,815 | 2,324,712 | 0.0312 |

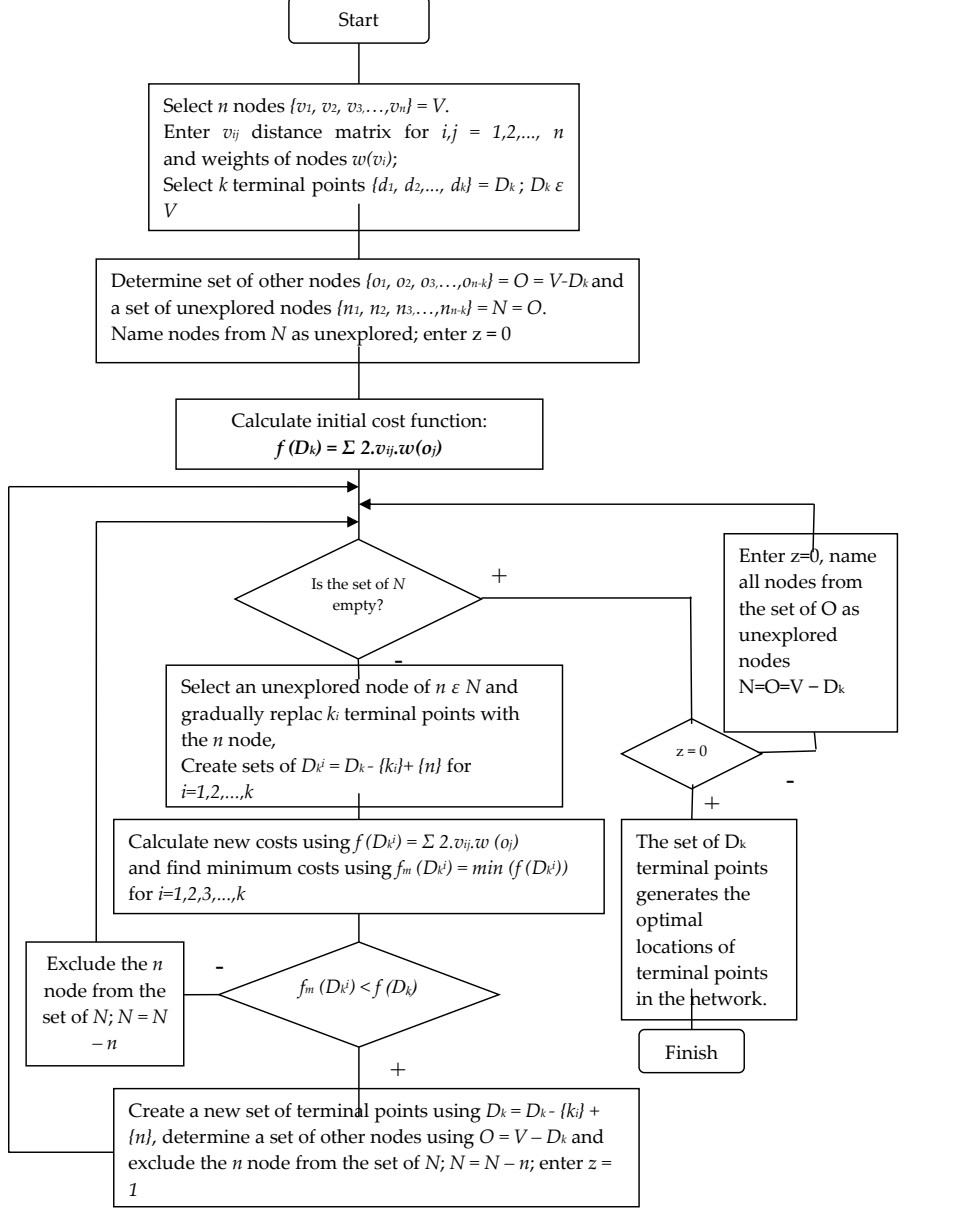

**Figure A1.** Iterative algorithm flowchart.

Table A2 and Table S2 contains a table with the initial matrix of objects. The matrix contains composite indicators for the individual vertices, according to which the weights of the individual vertices are determined in the mathematical model.

**Table A2.** Initial matrix of objects.

| Object | Potential of Combined Transport | C.T.T. | Rw.N. | Ro.N. | Composite Indicator |
|---|---|---|---|---|---|
| **Indicator Weights** | | 0.3 | 0.2 | 0.5 | |
| Bratislava | 202,495,806 | 1 | 1 | 1 | 404,991,612 |
| Malacky | 8,511,487 | 0 | 1 | 0 | 102,137,844 |
| Kúty | 1,653,254 | 0 | 1 | 1 | 2,479,881 |
| Holíč | 4,557,866 | 0 | 0 | 1 | 59,252,258 |
| Skalica | 5,909,698 | 0 | 0 | 0 | 5,909,698 |
| Senica | 831,194 | 0 | 0 | 1 | 10,805,522 |
| Pezinok | 1,004,036 | 0 | 1 | 0 | 12,048,432 |
| Senec | 7,142,347 | 0 | 1 | 1 | 107,135,205 |
| Dunajská Streda | 9,280,144 | 1 | 1 | 1 | 18,560,288 |
| Trnava | 27,420,268 | 0 | 1 | 1 | 41,130,402 |
| Galanta | 6,320,728 | 0 | 1 | 1 | 9,481,092 |
| Sereď | 6,832,232 | 0 | 1 | 1 | 10,248,348 |
| Hlohovec | 917,967 | 0 | 1 | 0 | 11,015,604 |
| Piešťany | 118,742 | 0 | 1 | 0 | 1,424,904 |
| Nové Mesto nad Váhom | 6,089,622 | 0 | 1 | 1 | 9,134,433 |
| Myjava | 3,750,362 | 0 | 0 | 1 | 48,754,706 |
| Trenčín | 16,555,924 | 0 | 1 | 0 | 198,671,088 |
| Dubnica nad Váhom | 747,054 | 0 | 1 | 1 | 1,120,581 |
| Ilava | 1,569,568 | 0 | 1 | 0 | 18,834,816 |
| Púchov | 5,410,482 | 0 | 1 | 1 | 8,115,723 |
| Považská Bystrica | 12,330,164 | 0 | 1 | 1 | 18,495,246 |
| Bytča | 3,248,865 | 0 | 1 | 1 | 48,732,975 |
| Žilina | 24,112,809 | 1 | 1 | 1 | 48,225,618 |
| Kysucké Nové Mesto | 4,673,025 | 0 | 1 | 0 | 560,763 |
| Čadca | 747,684 | 0 | 1 | 1 | 1,121,526 |
| Prievidza | 15,114,638 | 0 | 1 | 1 | 2,267,1957 |
| Partizánske | 71,687 | 0 | 1 | 0 | 860,244 |
| Topoľčany | 826,378 | 0 | 1 | 1 | 1,239,567 |
| Nitra | 2,468,562 | 0 | 1 | 1 | 4,937,124 |
| Šurany | 298,659 | 0 | 1 | 0 | 3,583,908 |
| Nové Zámky | 1,193,755 | 1 | 1 | 1 | 17,906,325 |
| Šaľa | 702,157 | 0 | 1 | 0 | 8,425,884 |
| Komárno | 1,054,557 | 0 | 1 | 1 | 15,818,355 |
| Hurbanovo | 230,822 | 0 | 1 | 0 | 2,769,864 |
| Štúrovo | 326,851 | 1 | 1 | 1 | 4,902,765 |
| Levice | 1,044,866 | 0 | 1 | 0 | 12,538,392 |
| Šahy | 1,737,524 | 0 | 1 | 1 | 2,606,286 |
| Vráble | 272,229 | 0 | 0 | 1 | 3,538,977 |
| Zlaté Moravce | 391,164 | 0 | 1 | 0 | 4,693,968 |
| Žarnovica | 1,409,958 | 0 | 1 | 1 | 2,114,937 |
| Žiar nad Hronom | 4,251,237 | 0 | 1 | 1 | 63,768,555 |
| Banská Štiavnica | 2,342,809 | 0 | 0 | 0 | 2,342,809 |
| Hontianske Nemce | 320,445 | 0 | 1 | 1 | 4,806,675 |
| Krupina | 1,687,677 | 0 | 1 | 0 | 20,252,124 |
| Zvolen | 9,364,115 | 0 | 1 | 1 | 140,461,725 |
| Hronská Dúbrava | 85,452 | 0 | 1 | 0 | 1,025,424 |
| Banská Bystrica | 17,652,959 | 0 | 1 | 1 | 264,794,385 |
| Turčianske Teplice | 191,828 | 0 | 0 | 0 | 191,828 |

**Table A2.** *Cont.*

| Object | Potential of Combined Transport | C.T.T. | Rw.N. | Ro.N. | Composite Indicator |
|---|---|---|---|---|---|
| **Indicator Weights** | | 0.3 | 0.2 | 0.5 | |
| Martin | 16,902,999 | 0 | 1 | 1 | 253,544,985 |
| Vrútky | 204,723 | 0 | 1 | 0 | 2,456,676 |
| Dolný Kubín | 5,616,531 | 0 | 1 | 1 | 84,247,965 |
| Kraľovany | 133,515 | 0 | 1 | 1 | 2,002,725 |
| Námestovo | 2,296,458 | 0 | 0 | 0 | 2,296,458 |
| Tvrdošín | 2,679,201 | 0 | 1 | 1 | 40,188,015 |
| Ružomberok | 8,527,158 | 1 | 1 | 1 | 17,054,316 |
| Liptovský Mikuláš | 9,328,248 | 0 | 1 | 0 | 111,938,976 |
| Brezno | 4,863,643 | 0 | 1 | 1 | 72,954,645 |
| Detva | 3,232,934 | 0 | 0 | 1 | 42,028,142 |
| Lučenec | 6,059,971 | 0 | 1 | 1 | 90,899,565 |
| Veľký Krtíš | 3,012,183 | 0 | 0 | 1 | 39,158,379 |
| Slovenské Ďarmoty | 113,936 | 0 | 0 | 0 | 113,936 |
| Fiľakovo | 2,214,631 | 0 | 1 | 0 | 26,575,572 |
| Rimavská Sobota | 534,075 | 0 | 0 | 1 | 6,942,975 |
| Poltár | 1,303,143 | 0 | 0 | 0 | 1,303,143 |
| Hnúšťa | 1,630,709 | 0 | 0 | 1 | 21,199,217 |
| Tornaľa | 1,730,403 | 0 | 1 | 1 | 25,956,045 |
| Rožňava | 4,573,536 | 0 | 1 | 1 | 6,860,304 |
| Revúca | 2,855,521 | 0 | 0 | 0 | 2,855,521 |
| Poprad | 11,966,306 | 0 | 1 | 1 | 17,949,459 |
| Spišská Nová Ves | 9,268,908 | 0 | 1 | 1 | 13,903,362 |
| Levoča | 3,121,969 | 0 | 0 | 0 | 3,121,969 |
| Spišský Štvrtok | 543,576 | 0 | 0 | 0 | 543,576 |
| Kežmarok | 274,9419 | 0 | 0 | 0 | 2,749,419 |
| Stará Ľubovňa | 3,524,323 | 0 | 0 | 1 | 45,816,199 |
| Sabinov | 2,652,556 | 0 | 1 | 0 | 3,1830,672 |
| Bardejov | 7,182,764 | 0 | 0 | 1 | 93,375,932 |
| Svidník | 2,667,458 | 0 | 0 | 1 | 34,676,954 |
| Prešov | 19,804,758 | 0 | 1 | 1 | 29,707,137 |
| Spišské Podhradie | 918,456 | 0 | 0 | 1 | 11,939,928 |
| Margecany | 487,344 | 0 | 1 | 1 | 731,016 |
| Gelnica | 1,490,148 | 0 | 1 | 0 | 17,881,776 |
| Moldava nad Bodvou | 2,314,884 | 0 | 1 | 0 | 27,778,608 |
| Košice | 56,232 | 1 | 1 | 1 | 112,464 |
| Trebišov | 543,576 | 0 | 1 | 0 | 6,522,912 |
| Slovenské Nové Mesto | 253,044 | 0 | 1 | 0 | 3,036,528 |
| Michalovce | 9,540,696 | 0 | 1 | 1 | 14,311,044 |
| Sobrance | 149,952 | 0 | 0 | 0 | 149,952 |
| Dobrá pri Č.n.T. | 9,372 | 1 | 1 | 0 | 159,324 |
| Vranov nad Topľou | 4,947,464 | 0 | 0 | 1 | 64,317,032 |
| Strážske | 1,068,408 | 0 | 1 | 1 | 160,2612 |
| Humenné | 7,532,961 | 0 | 1 | 1 | 112,994,415 |
| Snina | 4,597,267 | 0 | 0 | 0 | 4,597,267 |
| Medzilaborce | 1,438,043 | 0 | 1 | 0 | 17,256,516 |
| Stropkov | 2,324,712 | 0 | 0 | 0 | 2,324,712 |

Explanatory notes: C.T.T.—Container Trans-shipment Terminal. Rw.N.—Railway Node. Ro.N.—Road Node.

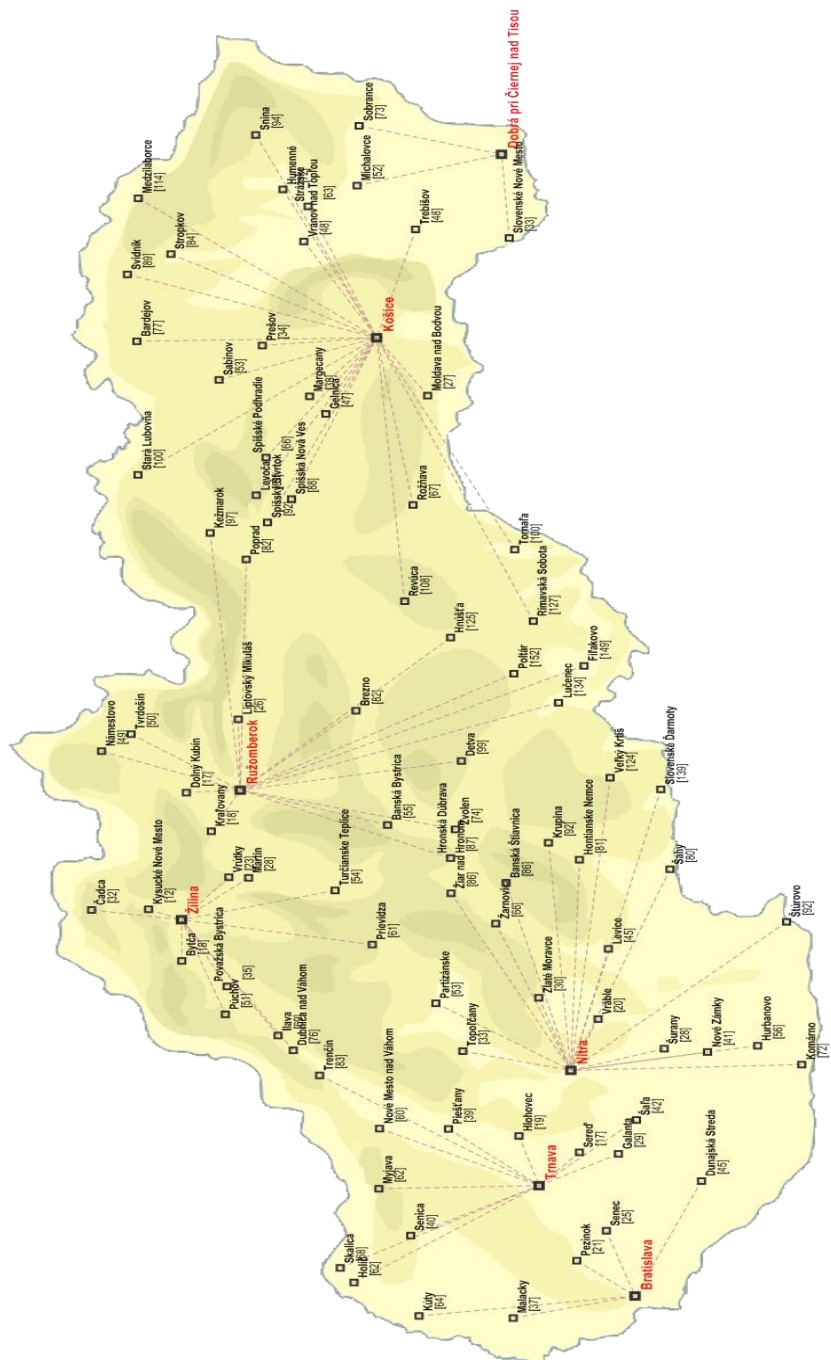

**Figure A2.** Contains a proposed locations of intermodal transport terminals in the Slovak Republic (SR). Source: Author.

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
