# Peer review of "Methodology for Determining the Location of Intermodal Transport Terminals for the Development of Sustainable Transport Systems: A Case Study from Slovakia"

_sustainability, doi:10.3390/su11051230_

Round 1

Reviewer 1 Report

Dear Author,

thank you for your new version of paper.

Regards,

Author Response

Dear Reviewer,

thank you very much.

Author

Reviewer 2 Report

The topic is interesting and the paper is well written and presented. The aim and research questions of the paper need to be clearly presented in the introduction part and the literature review needs to be complemented with more aspects of intermodal transport and the potential benefits from its development and further use. Sustainability aspects of intermodal transport are not analysed, so as to make clear why the development of intermodal transport leads to sustainable transport systems.

Author Response

Dear Reviewer,

thank you very much for your review.

The Introduction has been complemented by defining the objective of the contribution. There have been identified the links between a sustainable transport system and the development of intermodal transport.

The Literature Review has been extended by analysis of literature dealing with the issue of intermodal transport influence on sustainable transport system. The authors point to the environmental benefits of the intermodal transport and the need for cooperation between the individual modes of transport. The accessibility of the intermodal transport terminals is a necessary condition for the intermodal transport development.

Reviewer 3 Report

The presented method lacks a clarified methodical reference to the economic potential of the analyzed locations. The great importance of population for the assessment of node potential is an exaggerated simplification. The entire methodology refers to data of distribution terminals without taking into account other functions (e.g. gate terminals, dry terminals). Additionally, there is a need to take into account in the methodology the differences between the handling of various loading units [what has been noticed in Introduction] (swap bodies, semitrailers, air-cargo units) and various intermodal chains (rail-road, river-road, air-road). 

A big problem for the reader is the use of variable and non-consonantly names (economically neutral-commercial neutrality; vertex-node-object; composite indicator-vertex coefficient; etc.).

One remark more regarding the 150km  limit: tax exemptions are much less important for combined transport comparing to 44 tons and weekend-operations allowance.

Author Response

Dear Reviewer,

thank you very much for your review.

a) Chapter 4.1.1. explains the importance of intermodal transport as the main indicator for locating the terminals.  The potential of the intermodal transport is the basic economic indicator, as it expresses the volume of the goods flows in the location. This location is known from available statistics. However, the potential is expressed for a region, while we need to divide this potential into individual vertices. Perhaps the most accurate method is to define the vertex coefficient based on the gross domestic product. Gross domestic product is most relevant indicator of the “vertex economic strength”. However, the available statistics for the Slovak Republic do not give the GDP values for individual towns (vertices), but only for the individual regions of the Slovak Republic. In the case study of locating terminals in the SR territory, the author used the indicator “number of inhabitants”. In the case of another study (if the GDP statistic were known in more detail), the GDP would be used for the calculation of the coefficient.

b) In terms of methodology for locating intermodal transport terminals, the author does not considers the difference between the transhipment of a container, swap body or intermodal road semi-trailer. The author draws on the requirements for an intermodal transport terminal defined in the AGTC agreement. According to the agreement, a terminal has to be able to tranship any transport unit. The handling equipment thus has to be equipped with gripping device for handling containers, swap bodies or road semi-trailers. The type of transport unit does not have any influence on the terminal location.

c) As for other functions (warehouses, gate terminals) – terminal is always a central element of any bigger logistic centre. In the practice, it is clear that after constructing a terminal, warehouses and distribution centres are built in its immediate vicinity. The existence of a terminal (available transport infrastructure) causes a synergistic effect in the form of constructing other warehouses, thus extending the services provided by other logistic services. However, the methodology is focused on the substantial part of such centres – terminals, whose construction supports the development of sustainable transport systems in the territory.

d) The author has aligned the terminology in the contribution.

e) The author of the contribution shares the view that increasing the total weight of combination of vehicles or weekend roadblock exceptions are greater incentives but those cannot be included in the methodology proposed. Tax reliefs depend on the distance travelled by road; the author therefore decided to include this incentive in the methodology.   

Round 2

Reviewer 2 Report

The manuscript has been significantly improved and my comments have been sufficiently answered.

This manuscript is a resubmission of an earlier submission. The following is a list of the peer review reports and author responses from that submission.

Round 1

Reviewer 1 Report

Dear Author,

I find topic of your paper excellent. Paper is very well structured. Literature review is state of the art and comprehensive. As always there is some place for improving your paper. Here are my remarks:

1. Line 120 you have: [j1] – this is some technical error

2. Line 221 Va vertex – please check this notation

3. Line 229 – 301 On this point is necessary to explain weights. It’s the railway is the least influential? Or you can draw other conclusion?

4. Chapter Discussion is in this form is not so informative. Can you add more information about location of the terminal, there connection to railway and road infrastructure. Also, what is happing on the border between two terminals, for example: Trenčin and Dubnica nad Vahom. According to the distance Trnava and Nitra, Žilina and Ružoberok, are quite different, on this point it necessary comment about this issue.

5. In Conclusion – is there any still open question to raise up, for example future steps.

6. Appendix A – there is some technical issue with diagram – please check – for example “If the set of …”

7. Appendix B – please justified all numbers to right border – then it will be easier to read

8. Appendix C – see remarks under Appendix B

Regards,